# Intra-Rater Reliability of Shear Wave Elastography for the Quantification of Respiratory Muscles in Adolescent Athletes

**DOI:** 10.3390/s22176622

**Published:** 2022-09-01

**Authors:** Małgorzata Pałac, Paweł Linek

**Affiliations:** 1Musculoskeletal Elastography and Ultrasonography Laboratory, Institute of Physiotherapy and Health Sciences, The Jerzy Kukuczka Academy of Physical Education, 40-065 Katowice, Poland; 2Musculoskeletal Diagnostic and Physiotherapy—Research Team, The Jerzy Kukuczka Academy of Physical Education, 40-065 Katowice, Poland

**Keywords:** shear wave elastography, adolescent, athletes, diaphragm, intercostal muscles

## Abstract

The aim of this study was to assess the intra-rater reliability and agreement of diaphragm and intercostal muscle elasticity and thickness during tidal breathing. The diaphragm and intercostal muscle parameters were measured using shear wave elastography in adolescent athletes. To calculate intra-rater reliability, intraclass correlation coefficient (ICC) and Bland–Altman statistics were used. The reliability/agreement for one-day both muscle measurements (regardless of probe orientation) were at least moderate. During the seven-day interval between measurements, the reliability of a single measurement depended on the measured parameter, transducer orientation, respiratory phase, and muscle. Excellent reliability was found for diaphragm shear modulus at the peak of tidal expiration in transverse probe position (ICC_3.1_ = 0.91–0.96; ICC_3_._2_ = 0.95), and from poor to excellent reliability for the intercostal muscle thickness at the peak of tidal inspiration with the longitudinal probe position (ICC_3.1_ = 0.26–0.95; ICC_3_._2_ = 0.15). The overall reliability/agreement of the analysed data was higher for the diaphragm measurements (than the intercostal muscles) regardless of the respiratory phase and probe position. It is difficult to identify a more appropriate probe position to examine these muscles. The shear modulus/thickness of the diaphragm and intercostal muscles demonstrated good reliability/agreement so this appears to be a promising technique for their examination in athletes.

## 1. Introduction

Respiratory muscles are considered not only in the context of the respiratory system, but also in relation to spine stability, intra-abdominal pressure [1,2], pain sensation [3,4] and body balance [5]. Respiratory muscle morphology (mainly the diaphragm and intercostal muscles) can be assessed using ultrasound imaging (US) [6,7,8,9]. Recently, shear wave elastography (SWE) as a new, non-invasive US imaging technique, has allowed the assessment of muscle’s mechanical properties [10,11]. It has been suggested that SWE may be used as an index of diaphragmatic force change [12], and that the diaphragm shear modulus measured using SWE is related to transdiaphragmatic pressure [13,14], which is considered the gold standard in diaphragm evaluation.

There are a number of studies assessing the reliability of US diaphragm thickness [3,15,16], echogenicity [17], excursion [18,19,20] or velocity [21,22] measurements. The reliability of intercostal muscle US measurements were, in turn, evaluated only in four studies [7,23,24,25]. The reliability of the diaphragm SWE was only measured in two studies [26,27] on a limited population (healthy adults or chronic obstructive pulmonary disease and critically ill patients), whereas intercostal muscles were only analysed in one study [23]. To the best of our knowledge, there is only one reliability study on intercostal muscle SWE [23] and no study of diaphragm SWE in adolescents. Although, diaphragm and intercostal muscle SWE or thickness is usually measured in adults or patients with impaired respiratory system, it could be useful to assess the SWE of these muscles in adolescent athletes. The reliability results for the adults (who were sometimes critically ill) should not be transferred to healthy athletes in whom the functioning of the respiratory system (and respiratory muscles) function is expected to be at or above the population norm. It was confirmed that athletes have a greater diaphragm thickness [16] and higher pulmonary parameters than non-athletes [28,29].

Vicente-Campos et al. [3] have suggested that diaphragm exercise should be a crucial component of sports performance, injury prevention and rehabilitation strategy. Therefore, it is important to consider investigation of respiratory muscles (especially the diaphragm) in a broader (not just respiratory-related) context and on heterogeneous populations. As an example, there is a relationship between diaphragm thickness and non-specific lumbopelvic pain in athletes [9]. We believe that extensive research considering respiratory muscle measurements by SWE in adolescent athletes could provide new knowledge about the physiology of these muscles and potentially influence training, diagnostic, prognostic, or rehabilitation procedures. However, the reliability and agreement of SWE will be important to ensure the study and measurement quality in future studies assessing respiratory muscles in adolescent healthy athletes. Thus, the aim of this study was to assess the intra-rater reliability and agreement of diaphragm and intercostal muscle elasticity and thickness during tidal breathing.

## 2. Materials and Methods

### 2.1. Setting and Study Design

This study was conducted in Musculoskeletal Elastography and Ultrasonography Laboratory in accordance with the Declaration of Helsinki. The protocol was approved by local Ethics Committee (Decision No. 9/2020). All participants and their parents were informed about the procedures performed and provided written informed consent to participate in the study.

### 2.2. Participants

Ten male footballers (mean age: 17.1 ± 0.29; mean body mass: 71.5 ± 7.57; mean body height: 179.9 ± 5.67; BMI: 22.1 ± 1.90 kg/m^2^; football participation from 7 to 9 years) were selected using convenience sampling from an elite youth football club.

### 2.3. Investigator

Ultrasound data (SWE, thickness) were collected and analysed by a physiotherapist. Prior to the study, examiner had 3 years of experience in musculoskeletal SWE. Before the study, the examiner was additionally trained by an experienced radiologist in evaluating the respiratory muscles and underwent 3 months practical training.

### 2.4. Equipment

An Aixplorer ultrasound scanner (Product Version 12.2.0, Sofware Version 12.2.0.808, Supersonic Imagine, Aix-en-Provence, France) coupled with a linear transducer array (2–10 MHz; SuperLinear 10-2, Vermon, Tours, France) was used to evaluate the diaphragm and intercostal muscles’ shear modulus and thickness.

### 2.5. Measurement Procedures

The measurements were performed on the right body side in the supine position using SWE mode. The patient’s right hand was placed under the head in order to better visualize the diaphragm on the US. At the beginning, the examiner marked anterior and mid-axillary line on the chest, and positioned the US probe between them (the right intercostal space). The probe was positioned in the first intercostal space (counting from the bottom) where the lungs did not obscure the diaphragm during tidal breathing. The US measurements were performed in two probe orientations: transversally to the ribs—long body axis (Figure 1A) and parallel to the ribs—space between two ribs (Figure 1B).

The participants were asked to stay calm and breath quietly throughout the measurement procedure. US data was collected twice at the end-tidal inspiration and at the end-tidal expiration, separately. The moment of determining the end stage of inspiration and expiration was based on the visual inspection of diaphragm movement on real-time US. The end of diaphragm movement during tidal breathing was defined as the end of tidal inspiration or expiration.

After 7 days, the procedure was replicated in order to calculate reliability in a more extended time interval. The examiner was encouraged to apply minimum force by US probe to the skin because this may have affected the study results [30]. The left side was not examined due to the smaller acoustic window affecting reliability [20].

### 2.6. Data Analysis

From each US image collected in SWE, mode thickness and shear modulus (elasticity) measurements were collected. The Q-Box quantitative tool was used to quantify muscle shear modulus. Three circles were positioned in the middle of the image and inside the fascial edge of each muscle between the ribs. The circles were always next to each other and omitted potential artefacts (when they were detected).

In order to measure thickness precisely, the images were saved on an external drive in DICOM format and transferred to a computer where they were further processed using RadiAnt DICOM Viewer (Medixant, Poznań, Poland). If needed, images were sharpened, enlarged and contrasted to better visualize the pleural line and the peritoneal line. The diaphragm thickness was measured between these two hyperechoic lines. The intercostal muscles were measured as the first muscle placed was more superficial than the diaphragm (Figure 2). Shear modulus and thickness of the muscles were measured manually based solely on the examiner‘s experience.

### 2.7. Statistical Analyses

To calculate intra-rater reliability, intraclass correlation coefficient (ICC) type 3.1 (for single measurement) and type 3.2 (for mean value from two measurements) were used. The ICC was interpreted according to the following criteria: 1.00–0.75 (excellent), 0.74–0.60 (moderate), 0.59–0.40 (fair), and below 0.40 (poor reliability) [31]. In order to calculate agreement, the standard error of measurement (SEM = SD ×$\;\sqrt{1-\mathrm{ICC}}$), the coefficient of variation (CV), and the results of the Bland–Altman test (BA) were used. The only reason to use the BA test was to find potential biases between the two measures. Due to the sample size not being large enough (more than 50 is preferred to allow a good estimation of the limits of agreement), plots with limits of agreement were not included [32]. The significance level was set at *p* < 0.05. Data were analysed using STATISTICA 13.1 PL (Statsoft, Tulsa, OK, USA) and Excel 2013 (Microsoft Corporation, Redmond, Washington, DC, USA) software.

## 3. Results

### 3.1. Transverse Probe Orientation (Transversally to the Ribs)

The one-day intra-session reliability (ICC_3.1_) of diaphragm and intercostal muscles shear modulus at peak of tidal expiration and inspiration was generally excellent. The corresponding CV was always below 3% at inspiration phase and below 1% at expiration phase; no systematic errors in BA test were detected. The intra-session reliability for single measurement (ICC_3.1_) during the 7-day interval varied from excellent to moderate for diaphragm and from fair to poor for intercostal muscles. The intra-session reliability (ICC_3.2_) for the mean value from two measurements was improved (excellent for diaphragm, fair to excellent for intercostal muscles). Corresponding CV was always below 8%, but systematic error was detected for diaphragm shear modulus at peak tidal inspiration. The expiration phase always corresponds with higher ICC and lower SEM and CV.

The diaphragm and intercostal muscle thickness demonstrated excellent one-day intra-session reliability (ICC_3.1_) with CV below 9%. The BA test showed negative mean bias with systematic error for intercostal muscles at peak inspiration and expiration and positive bias for diaphragm. The intra-session reliability for single measurement (ICC_3.1_) during the 7-day interval varied from excellent to fair, but no systematic errors were detected. The intra-session reliability (ICC_3.2_) for the mean value from the two measurements varied from excellent to moderate, and the corresponding CV did not exceed 6.03%.

The mean bias was close to zero for the diaphragm and intercostal muscle thickness measurements during peak tidal inspiration and expiration. All results of the reliability and variability in transverse probe orientation are included in Table 1.

### 3.2. Longitudinal Probe Orientation (Parallel to the Ribs)

The one-day intra-session reliability (ICC_3.1_) of the diaphragm and intercostal muscles’ shear modulus at peak of tidal expiration and inspiration varied from excellent to moderate. Corresponding CV was always below 4%. The intra-session reliability for single measurement (ICC_3.1_) during the 7-day interval varied from fair to poor but the corresponding CV was always below 4%. The intra-session reliability (ICC_3.2_) for the shear modulus mean value from two measurements was improved (moderate for all measurements) and the CV was still below 4%. In the longitudinal probe orientation, the BA test showed bias below 2 kPa with no systematic errors. The expiration and inspiration phases showed similar reliability and agreement results.

The diaphragm and intercostal muscles’ one-day intra-session reliability (ICC_3.1_) of thickness measurements varied from moderate–excellent with CV below 10%. The BA test showed a mean bias close to zero with no systematic errors. The intra-session reliability for single measurement (ICC_3.1_) during the 7-day interval was moderate for the diaphragm and varied from poor to fair for the intercostal muscles. The corresponding CV was always below 6%. The intra-session reliability (ICC_3.2_) for the mean value from two thickness measurements was improved with the exception of the intercostal muscles during inspiration. The corresponding CV did not exceed 6.37% and no systematic errors were detected in any cases. All results of reliability and variability in longitudinal probe orientation are included in Table 2.

## 4. Discussion

The main aim of the study was to assess the reliability and agreement of shear modulus measurements in diaphragm and intercostal muscles at peak of tidal expiration and inspiration. To the best of our knowledge, no other studies have calculated the reliability and agreement of these muscle measurements in adolescent athletes. In our study, we showed that regardless of the probe orientation and the muscle tested, the reliability/agreement for one-day measurements were at least moderate. However, at transverse probe positioning, a bias in the thickness measurements of the diaphragm and intercostal muscle was detected (in the second measurement, higher values were found compared to the first measurement). From a clinical perspective, it is more reasonable to analyse reliability/agreement at longer intervals. During the seven-day interval between measurements, the reliability of a single measurement depended on the measured parameter, transducer orientation, respiratory phase, and muscle. Excellent reliability was found for diaphragm shear modulus at the peak of tidal expiration in transverse probe position, and poor reliability for intercostal muscle thickness at the peak of tidal inspiration with the longitudinal probe position. At the 7-day interval, the analysis of the mean values from the two measurements allowed moderate reliability for almost all variables analysed (with the exception being the reliability of intercostal muscle thickness at the peak of tidal inspiration in the longitudinal probe position), and the CV for all variables was remarkably below 10%. The overall reliability/agreement of the analysed data was higher for the diaphragm elasticity and thickness measurements (in relation to the intercostal muscles) regardless of the respiratory phase and probe position. The longitudinal probe position is characterised by a lack of bias and slightly lower CV values.

In the literature, there are a number of studies assessing the reliability of diaphragm and intercostal muscle thickness in adults with diseases [7,27,33], healthy adults [19,25] or athletes [3,15]. Out of these, there were only two studies evaluating the diaphragm thickness reliability in adolescents [34,35], where ICC or bias at the peak of tidal expiration was similar to the present study’s results [34,35]. These results were similar despite the use of a different methodology, a larger age span, and the use of a different interval between repeated measurements. In studies on adults, the reliability of diaphragm thickness at the end of maximal inspiration [18,36,37,38,39,40], at the end of tidal expiration [3,15,18] and at the end of tidal inspiration [18,39,41] was confirmed. In turn, the reliability of intercostal muscles thickness at the peak of tidal expiration [7,24,25] and at the end of maximal inspiration ranged from 0.6 to 0.9 [24,25], which was also consistent with the results obtained in this study.

The reliability of diaphragm shear modulus was only assessed in adults [26,27], and intercostal muscles in adolescents as well [23]. The intra-rater ICC for diaphragm elasticity was excellent for measurements at the end of tidal expiration [26] and at apnea after expiration [27]. The ICC for intercostal muscles was also excellent during normal breathing and in apnea [23]. In all of these studies, the reliability was calculated for data collected during the same day, and was similar to the present study results (the one-day reliability of the diaphragm and intercostal muscle elasticity was also excellent in the transverse probe position).

In the present study, we also attempted to determine the reliability/agreement of the intercostal muscles and diaphragm US measurements, taking into account the probe orientation (transverse vs. longitudinal). This is particularly important in the assessment of elasticity by SWE, as evidence shows that the probe orientation in relation to muscular fibres may affect the results [42]. The diaphragm shear modulus reliability was evaluated in longitudinal [26,27] and transverse [26] probe orientation, but intercostal muscle shear modulus was only analysed in the transverse probe orientation [23]. In all of these studies, the reliability for one-day measurements was excellent. Only in the study by Flattres et al. [26] was the obtained reliability poor for the diaphragm assessment in the transverse probe orientation. In our study, longitudinal and transverse probe orientation resulted in excellent reliability for assessing diaphragm and intercostal muscle elasticity during tidal inspiration. During expiration, we found better reliability in the transverse probe orientation, which is only contrary to the results obtained in the study by Flattres et al. [26], where better reliability was observed in the longitudinal probe position. This may be due to differences in the population studied. In our study, there were slender adolescent athletes, whereas in the study by Flatters et al. [26] adults were recruited. Regular sporting activity influences the lungs and chest elasticity [28], and may explain the differences in reliability of the elasticity measurements between longitudinal and transverse probe position.

The present study had a number of limitations. First, the sample size was small and homogenous (adolescent athletes), and the results should be applied with caution to different populations. Second, the examiner had relatively little experience in the diaphragm and intercostal muscle assessment. However, SWE does not require much examiner experience in assessing the diaphragm [26]. In the present study, one-day reliability was excellent for most of the variables analysed. Third, probe compression was not controlled by an external device or specialised US gel pad. Another study showed that probe stabilizing grips may affect the muscle‘s elasticity [30]. Fourth, in the present study, we evaluated only intra-examiner reliability and an inter-examiner calculation needs to be performed. Fifth, the athletes were only measured in supine position. It is frequent practice to examine the diaphragm in other body positions (e.g., semi-supine, seated)—so it is worth remembering that the reliability values in the present study (and other work cited) only apply to the supine position (body position can affect diaphragm relaxation). Six, measurements were only collected during tidal breathing.

## 5. Conclusions

Shear modulus/thickness of the diaphragm and intercostal muscles during tidal breathing demonstrated good reliability/agreement in adolescent athletes. However, the diaphragm had better reliability. SWE appears to be a promising technique to examine the diaphragm and intercostal muscles in athletes. At this stage, it is difficult to unambiguously identify a more appropriate probe position (transverse vs. longitudinal). We currently recommend taking at least two repeated measurements and analysing the mean value. Further studies are needed to establish an optimal measurement procedure and improve the reliability (in particular during intercostal muscle assessment at tidal inspiration).

## Figures and Tables

**Figure 1 sensors-22-06622-f001:**
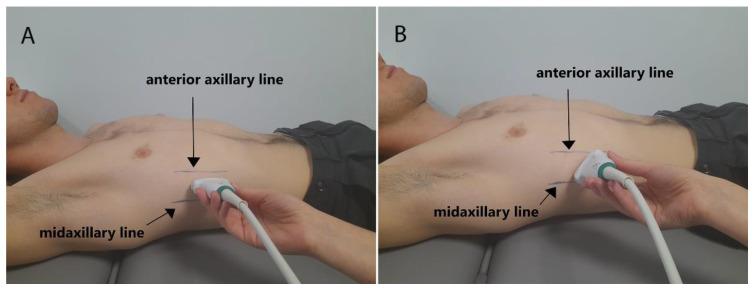
Illustration showing the placement of the transversally (**A**) and parallel to the ribs (**B**) ultrasound probe position.

**Figure 2 sensors-22-06622-f002:**
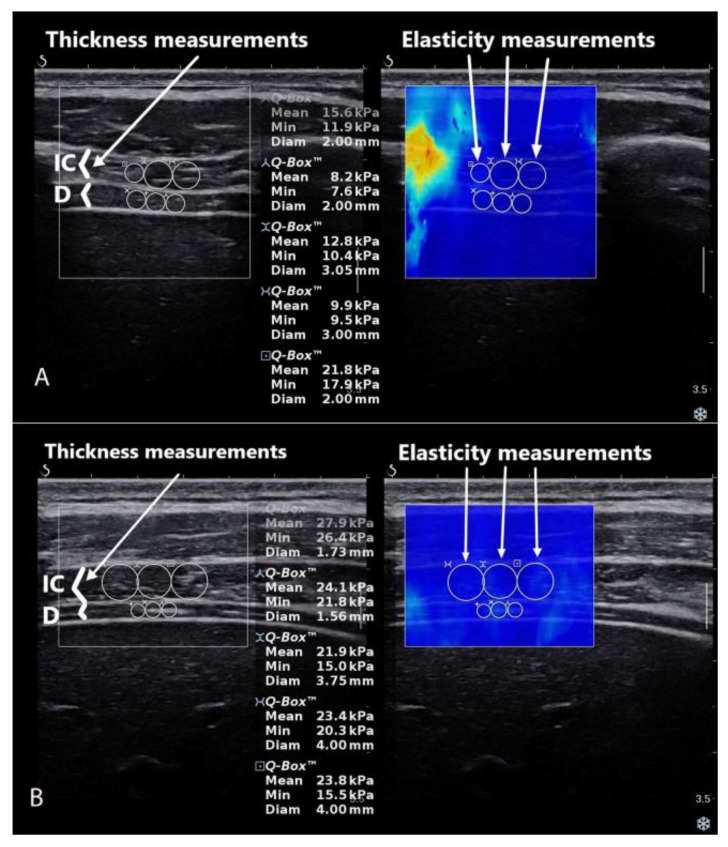
Data extraction from the images collected in SWE mode in transverse (**A**) and longitudinal (**B**) probe view. D—diaphragm; IC—intercostal muscle.

**Table 1 sensors-22-06622-t001:** Reliability and validity of stiffness and thickness values measured in diaphragm (D) and intercostal (IC) muscles in transverse probe orientation.

		Inspiration		Expiration	
		D	IC	D	IC
**Shear modulus**	Mean (kPa) ^1^	27.36	24.96	25.42	23.81
	SD (kPa) ^1^	5.29	4.92	5.31	5.20
**Intra-session reliability (1 day)**	ICC_3_._1_	0.87	0.86	0.96	0.95
	SEM (kPa)	2.13	1.81	1.07	1.03
	CV (%)	2.98	1.60	0.71	0.17
	Bias ^3^ (kPa)	1.22	0.58	−0.26	0.06
**Intra-session reliability (after 7 days)**	ICC_3_._1_	0.66	0.35	0.91	0.51
	SEM (kPa)	3.40	4.00	1.61	3.68
	CV (%)	10.63	5.69	3.14	0.54
	Bias ^3^ (kPa)	4.14^2^	1.99	1.13	−0.18
	ICC_3_._2_	0.78	0.47	0.95	0.76
	SEM (kPa)	2.53	3.66	1.18	2.53
	CV (%)	7.32	3.36	2.94	0.50
	Bias ^3^ (kPa)	3.34 ^2^	1.09	1.28	−0.07
**Muscle thickness**	Mean (mm) ^1^	1.79	3.65	1.48	3.71
	SD (mm) ^1^	0.62	0.74	0.32	0.88
**Intra-session reliability (1 day)**	ICC_3_._1_	0.93	0.76	0.80	0.93
	SEM (mm)	0.20	0.43	0.15	0.25
	CV (%)	2.21	8.66	6.37	4.48
	Bias ^3^ (mm)	0.06	−0.44 ^2^	0.14 ^2^	−0.23 ^2^
**Intra-session reliability (after 7 days)**	ICC_3_._1_	0.80	0.51	0.48	0.84
	SEM (mm)	0.29	0.54	0.25	0.36
	CV (%)	0.00	8.11	6.55	3.72
	Bias ^3^ (mm)	−0.001	−0.41	0.14	−0.19
	ICC_3_._2_	0.85	0.65	0.75	0.92
	SEM (mm)	0.24	0.46	0.17	0.25
	CV (%)	1.48	6.03	4.91	2.75
	Bias ^3^ (mm)	−0.01	−0.04	0.08	−0.06

CV—coefficient of variation; ICC—intraclass correlation coefficient; SEM—standard error of the mean; ^1^ from all (four) measurements; ^2^ systematic error as the line of equality is not in the 95% confidence interval; ^3^ Bland–Altman test.

**Table 2 sensors-22-06622-t002:** Reliability and validity of stiffness and thickness values measured in diaphragm (D) and intercostal (IC) muscles in longitudinal probe orientation.

		Inspiration		Expiration	
		D	IC	D	IC
**Shear modulus**	Mean (kPa) ^1^	30.69	26.66	28.60	25.94
	SD (kPa) ^1^	6.36	6.26	7.15	5.72
**Intra-session reliability (1 day)**	ICC_3_._1_	0.94	0.85	0.68	0.44
	SEM (kPa)	1.27	1.99	3.85	4.09
	CV (%)	0.41	3.18	2.03	3.44
	Bias ^2^ (kPa)	−0.18	1.20	0.83	1.24
**Intra-session reliability (after 7 days)**	ICC_3_._1_	0.43	0.50	0.38	0.52
	SEM (kPa)	5.33	4.52	5.68	3.90
	CV (%)	0.23	1.04	3.08	0.12
	Bias ^2^ (kPa)	0.10	0.40	1.25	0.04
	ICC_3_._2_	0.63	0.65	0.65	0.66
	SEM (kPa)	3.92	3.74	4.33	3.52
	CV (%)	3.14	2.10	1.84	2.58
	Bias ^2^ (kPa)	1.00	0.15	0.63	−0.72
**Muscle thickness**	Mean (mm) ^1^	1.83	3.81	1.56	3.85
	SD (mm) ^1^	0.61	0.68	0.42	0.83
**Intra-session reliability (1 day)**	ICC_3_._1_	0.96	0.95	0.68	0.84
	SEM (mm)	0.15	0.19	0.29	0.39
	CV (%)	0.29	1.18	9.41	2.18
	Bias ^2^ (mm)	−0.007	−0.06	0.21	−0.12
**Intra-session reliability (after 7 days)**	ICC_3_._1_	0.67	0.26	0.73	0.59
	SEM (mm)	0.33	0.68	0.27	0.55
	CV (%)	1.58	0.35	5.13	2.87
	Bias ^2^ (mm)	−0.04	−0.02	0.12	0.15
	ICC_3_._2_	0.85	0.15	0.78	0.64
	SEM (mm)	0.24	0.67	0.21	0.51
	CV (%)	1.69	1.69	6.37	3.57
	Bias ^2^ (mm)	−0.05	−0.05	0.06	0.22

CV—coefficient of variation; ICC—intraclass correlation coefficient; SEM—standard error of the mean; ^1^ from all (four) measurements; ^2^ Bland–Altman test.

## Data Availability

The datasets generated during and/or analysed during the current study are available from the first or corresponding author on reasonable request.

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
