# Peer review of "Intra-Rater Reliability of Shear Wave Elastography for the Quantification of Respiratory Muscles in Adolescent Athletes"

_sensors, 2022, doi:10.3390/s22176622_

Round 1
Reviewer 1 Report
The article "Intra-rater reliability of shear wave elastography for the quantification of respiratory muscles in adolescent athletes" submitted for review presents an interesting and relevant topic of using shear wave elastography. As the reviewed article was proposed for publication in the section "Sensing-Based Biomedical Communication and Intelligent Identification for Healthcare" Sensors, my comments mainly concern the measurement techniques applied and the data processing used for analysis.
1. „ The diaphragm thickness was measured from the pleural line to the peritoneal line”
How were the edges of these lines delineated in the images to minimize errors? Was there an image processing algorithm or a baseline based solely on the operator's experience?
What were the criteria for the distribution of the circles? Was it manually determined by the operator or was an image processing algorithm used?
These aspects should be commented on in the manuscript.
2. “US data was collected twice at the end-tidal inspiration and at the end-tidal expiration, separately”.
How were the data acquisition moments (breath extremes) determined? Please explain why a reference respiratory phase measuring system was not used. In my opinion, the method of measuring breathing should be described in the text.
If the moments of determining the extremes of respiratory activity were determined only on the basis of the operator's observations, it should be described, because it introduces a certain subjectivity into the conducted research.
3. In the article, the authors mention that "The examiner was encouraged to apply minimum force by US probe to the skin because this may affects the study results [31]." – How was the "minimum force by US probe" estimated? In Section 4 the authors indicate "... probe compression was not controlled by an external device or specialized US gel pad. An another study shown that probe stabilizing grips may affect the muscle`s elasticity [31]." Estimating the pressure force of the US head during the measurement would introduce greater objectivity into the obtained results. Rapid prototyping methods allow for the implementation of sensor solutions that can be integrated non-invasively with the scanning head, and which at the same time will not interfere with the measurement.
4. The authors used the model (3,1) and (3,2) to calculate the ICC. What is the rationale for using the model (3,2)? It should be explained why k=2, since the measurements were performed by one operator. In my opinion, this justification is missing.
Additionally, in Section 4, the authors indicate that "… the examiner had relatively little experience in the diaphragm and intercostal muscles assessment". So, how were the results obtained by one operator verified?
The authors claim that the measurements were made twice (on the first and seventh day), but you can get the impression that they were repeated many times for seven days (in Table 1 it says "intra-session reliability (7 days)"), it should rather be "the seventh day" or "after seven days".
Author Response
Dear Reviewer,
We appreciate the time and effort that you have dedicated to review our manuscript. We would also like to thank you for all your comments. We hope that the present form of the paper meets your expectation.
The article "Intra-rater reliability of shear wave elastography for the quantification of respiratory muscles in adolescent athletes" submitted for review presents an interesting and relevant topic of using shear wave elastography. As the reviewed article was proposed for publication in the section "Sensing-Based Biomedical Communication and Intelligent Identification for Healthcare" Sensors, my comments mainly concern the measurement techniques applied and the data processing used for analysis.
- „ The diaphragm thickness was measured from the pleural line to the peritoneal line”
How were the edges of these lines delineated in the images to minimize errors? Was there an image processing algorithm or a baseline based solely on the operator's experience?
RESPONSE: Thank you for this remark. In fact, pleural line and pertinoeal line are clearly visible as two hyperechoic (bright) lines on ultrasound images. The measurement of the diaphragm thickness between the pleural line and pertinoeal line was based on previous studies (Cappellini et al. 2012, Brown et al. 2013, Boon et al. 2013). We did not use any image processing algorithm and measurements were performed manually in order to reflect clinical practice. However, we have used RadiAnt DICOM Viewer to better visualize the pleural line and the peritoneal line.
In the present form of the study the following information was added:
If needed images were sharpen, enlarged and contrasted to better visualize the pleural line and the peritoneal line. The diaphragm thickness was measured between these two hyperechoic lines. The intercostal muscles were measured as the first muscle placed more superficial than diaphragm (Figure 2). Shear modulus and thickness of the muscles were measured manually based solely on the examiner`s experience.
What were the criteria for the distribution of the circles? Was it manually determined by the operator or was an image processing algorithm used?
RESPONSE: Of course yes. Thank you for this remark. We have added the following sentences:
Three circles were positioned in the middle of the image and inside the fascial edge of each muscle between the ribs. The circles were always next to each other and omitted potential artefacts (when was detected).
- “US data was collected twice at the end-tidal inspiration and at the end-tidal expiration, separately”.
How were the data acquisition moments (breath extremes) determined? Please explain why a reference respiratory phase measuring system was not used. In my opinion, the method of measuring breathing should be described in the text.
If the moments of determining the extremes of respiratory activity were determined only on the basis of the operator's observations, it should be described, because it introduces a certain subjectivity into the conducted research.
RESPONSE: Thank you for this remark. In fact, we omitted very important issue. The breathing phases can be easily detected by analysing the diaphragm movement on ultrasound. Thus, any other systems are not needed here. We have added two sentences to clarify this aspect.
The participants were asked to stay calm and breath quietly throughout the measurement procedure. US data was collected twice at the end-tidal inspiration and at the end-tidal expiration, separately. The moment of determining the end stage of inspiration and expiration was based on the visual inspection of diaphragm movement on real-time US. The end of diaphragm movement during tidal breathing was defined as the end of tidal inspiration or expiration.
- In the article, the authors mention that "The examiner was encouraged to apply minimum force by US probe to the skin because this may affects the study results [31]." – How was the "minimum force by US probe" estimated? In Section 4 the authors indicate "... probe compression was not controlled by an external device or specialized US gel pad. An another study shown that probe stabilizing grips may affect the muscle`s elasticity [31]." Estimating the pressure force of the US head during the measurement would introduce greater objectivity into the obtained results. Rapid prototyping methods allow for the implementation of sensor solutions that can be integrated non-invasively with the scanning head, and which at the same time will not interfere with the measurement.
RESPONSE: Thank you for your valuable advice. In future studies, we will consider testing the respiratory muscles under controlled pressure force by an external device. However, the purpose of this study was to check the reliability of US respiratory muscles measurements without external devices in order to reflect more real clinical environment and evaluate utility of US (without any external devices).
- The authors used the model (3,1) and (3,2) to calculate the ICC. What is the rationale for using the model (3,2)? It should be explained why k=2, since the measurements were performed by one operator. In my opinion, this justification is missing.
RESPONSE: In model 3 of ICC, the k means repeated measurements. In model 2 of ICC, the k means number of raters. Thus, in our study, the ICC type 3,1 means that the reliability was for a single measurement, whereas the ICC type 3,2 means that the reliability was calculated using mean value from two measurements on two different occasions.
For clarity, we have added the following sentence:
To calculate intra-rater reliability, intraclass correlation coefficient (ICC) type 3,1 (for single measurement) and type 3,2 (for mean value from two measurements) were used.
Additionally, in Section 4, the authors indicate that "… the examiner had relatively little experience in the diaphragm and intercostal muscles assessment". So, how were the results obtained by one operator verified?
RESPONSE: Thank you for this remark. In our study ”…the examiner had 3 years of experience in musculoskeletal SWE. Before the study, the examiner was additionally trained by an experienced radiologist in evaluating the respiratory muscles and underwent 3 months practical training” – see section 2.3. Thus, the examiner was experience in MSK SWE, but have relatively little experience in respiratory muscle assessment. However, in spite of this little experience the reliability was good. This confirm other study observation [26].
The authors claim that the measurements were made twice (on the first and seventh day), but you can get the impression that they were repeated many times for seven days (in Table 1 it says "intra-session reliability (7 days)"), it should rather be "the seventh day" or "after seven days".
RESPONSE: Thank you, you’re right. We changed it through the tables.
Reviewer 2 Report
Thank you for the opportunity to review this manuscript.
1) The introduction has not made it adequately clear to me why you need to assess the reliability in athletes instead of healthy adults (which as you say has already been done).
2) 10 is a small sample size. But you have already stated that as a limitation.
3) Did you assess reliability between US operators? If not, this is a limitation.
Author Response
Dear Reviewer,
We appreciate the time and effort that you have dedicated to review our manuscript. We would also like to thank you for all your comments. We hope that the present form of the paper meets your expectation
Thank you for the opportunity to review this manuscript.
- The introduction has not made it adequately clear to me why you need to assess the reliability in athletes instead of healthy adults (which as you say has already been done).
RESPONSE: Thank you for this remark. We have tried to explain this in the introduction:
“The reliability results for adults (who were sometimes critically ill) should not be transferred to healthy athletes in which the functioning of the respiratory system (and respiratory muscles) function is expected to be at or above the population norm. It was confirmed that athletes have a greater diaphragm thickness [28] and higher pulmonary parameters than non-athletes [29,30].”
In addition, we wanted to emphasize that the study of respiratory muscles in athletes is important because of its value:
“Respiratory muscles are considered not only in the context of the respiratory system, but also in relation to spine stability, intra-abdominal pressure [1,2], pain sensation [3,4] and body balance [5].”
“Vicente-Campos et al. [3] have suggested that diaphragm exercise should be a crucial component of sports performance, injury prevention and rehabilitation strategy. Therefore, it is important to consider investigation of respiratory muscles (especially the diaphragm) in a broader (not just respiratory-related) context and on heterogeneous populations. As an example, there is a relationship between diaphragm thickness and non-specific lum-bopelvic pain in athletes [9]. We believe that extensive researches considering respiratory muscles measurements by SWE in adolescent athletes could provide new knowledge about the physiology of these muscles and potentially influence training, diagnostic, prognostic, or rehabilitation procedures.”
- 10 is a small sample size. But you have already stated that as a limitation.
RESPONSE: Yes, we understand that this is a small group. In future research, we should consider a larger group and from different sports.
- Did you assess reliability between US operators? If not, this is a limitation.
RESPONSE: We did not access reliability between two operators. We have written it already in the paragraph related to limitations:
“Fourth, in the present study, we evaluated only intra-examiner reliability and an inter-examiner calculation needs to be performed.”
Round 2
Reviewer 1 Report
No comments.
Reviewer 2 Report
Thank you for adding in my previous comments. I feel that the methodology of this paper could be significantly improved, however, this could be future research. As such, I see no reason that this can not be accepted.